# Comparison of the Phenotypic Performance, Molecular Diversity, and Proteomics in Transgenic Rice

**DOI:** 10.3390/plants12010156

**Published:** 2022-12-29

**Authors:** Yue Sun, Huan Zhao, Zhongkai Chen, Huizhen Chen, Bai Li, Chunlei Wang, Xiaoli Lin, Yicong Cai, Dahu Zhou, Linjuan Ouyang, Changlan Zhu, Haohua He, Xiaosong Peng

**Affiliations:** 1Key Laboratory of Crop Physiology, Ecology and Genetic Breeding, Ministry of Education/College of Agronomy, JAU, Nanchang 330045, China; 2Jiangxi Biotech Vocational College, JAU, Nanchang 330200, China

**Keywords:** proteomics, transgenic rice, *Chilo suppressalis*, genetic background, unintended effect

## Abstract

The extent of molecular diversity and differentially expressed proteins (DEPs) in transgenic lines provide valuable information to understand the phenotypic performance of transgenic crops compared with their parents. Here, we compared the differences in the phenotypic variation of twelve agronomic and end-use quality traits, the extent of microsatellite diversity, and DEPs of a recurrent parent line with three transgenic rice restorer lines carrying either *CRY1C* gene on chromosome 11 or *CRY2A* gene on chromosome 12 or both genes. The three transgenic lines had significantly smaller stem borer infestation than the recurrent parent without showing significant differences among most agronomic traits, yield components, and end-use quality traits. Using 512 microsatellite markers, the three transgenic lines inherited 2.9–4.3% of the Minghui 63 donor genome and 96.3–97.1% of the CH891 recurrent parent genome. As compared with the recurrent parent, the number of upregulated and down-regulated proteins in the three transgenic lines varied from 169 to 239 and from 131 to 199, respectively. Most DEPs were associated with the secondary metabolites biosynthesis transport and catabolism, carbohydrate transport and metabolism, post-translational modification, and signal transduction mechanisms. Although several differentially expressed proteins were observed between transgenic rice and its recurrent parent, the differences may not have been associated with grain yield and most other phenotypic traits in transgenic rice.

## 1. Introduction

Rice is one of the major stable food crops for nearly half of the world’s population [1]. Rice production is affected by multiple insect pests, including the rice striped stem borer (*Chilo suppressalis*), rice gall midges, and rice thrips [2,3]. *C. suppressalis* is a chewing insect pest that causes significant physical damage to rice plants that causes substantial yield losses [4]. *C. suppressalis* larvae bore into the stems of rice plants and fed inside, resulting in “deadheart” at the tillering stage and “whiteheads” at the heading stage [5]. Transgenic technology has been widely used in crops to introduce foreign genes that regulate insect resistance, disease resistance, herbicide resistance, and heavy metal tolerance [6]. Bt insecticidal proteins produced by *Bacillus thuringiensis* have been widely used to genetically modify (GM) several food crops, including rice, corn, cotton, and other crops worldwide [2]. Bt produces crystalline (Cry) proteins, vegetative insecticidal (Vips) proteins, and cytolytic (Cyt) proteins, of which both Cry, and Vips proteins have been used to control insect pests. To date, research on Bt rice in China has mainly focused on breeding new lines, introducing *CRY1* (*CRY1Ab/Ac*, *CRY1C*), and *CRY2A* genes into indica rice cytoplasmic male sterility (CMS) restorer lines using transgenic technology [7]. Transgenic rice with *CRY1* and *CRY2* genes has shown obvious yield advantages under serious insect infestation as compared with varieties developed through conventional breeding methods [8].

The environmental and food safety of GM crops has been a major restricting factor in the commercialization of GM crops in many countries, including China [9], due to health concerns [3]. Potential sources of concern caused by transgenic technology can be divided into intended and unintended effects [10]. The intended effects of transgenic plants refer to the introduction of foreign genes into the plant genome to enhance or improve plant functions. In such cases, the introduced foreign genes are likely stably expressed in the recipient plants with the desired traits [11], which can be determined using nucleic acid sequences and gene expression analysis [12]. Real-time polymerase chain reaction (PCR) is one of the commonly used methods for molecular detection of nucleic acid sequences [13], whereas enzyme-linked immunosorbent assay (ELISA) is widely used to detect protein expressions [14]. Linkage drag is the unintended effect of transgenic crops, which simply refers to a reduction in the performance of lines or cultivars due to deleterious genes introduced along with the beneficial gene [15]. Unintended effects are difficult to detect, raising caution when assessing the risk of transgenic technology [16]. Therefore, unintended effects have been investigated using diverse methods that aim to compare the phenotypic performance and molecular variation of newly developed transgenic lines with their counterparts from parental or near-isogenic lines [17,18,19].

Researchers have studied the unintended effects of transgenic corn, rice, soybean, and other crops at the molecular level using transcriptomics, proteomics, and metabolomics [20,21,22,23,24,25,26], which revealed some differences between the transgenes and their parents [16,27,28]. The differences in the levels of transcriptomics, proteomics, and metabolomics between the transgenic and parental lines are likely caused by gene induction, insertion, transformation, and recombination [29,30]. Therefore, the precise identification of proteins is necessary to better understand the unintended effects of transgenic crops using 3D proteomics, 4D proteomics, and ion mobility, which separates ions mainly according to their shape and cross-section and can distinguish peptides with small M/Z difference, enabling the detection of low abundance protein signals [31]. It is particularly important to conduct bioinformatic analysis of clean proteome data, among which DEPs and COG are two key steps. The proteins that make up each COG are assumed to be derived from an ancestral protein. Orthologs are proteins from different species that have evolved from vertical lineages (speciation) and typically retain the same function as the original proteins. Previous studies of unintended effects generally conducted COG analysis through screened differentially expressed proteins, which can represent conserved pathways of biological processes [32,33]. The two common methods used to detect the expression level of foreign genes are real-time PCR and enzyme-linked immunosorbent assays (ELISAs) [34]. Real-time PCR detects non-expressing gene sequences, while ELISA requires gene expression to detect the presence of a foreign gene product (protein). The genetic background of the transgenic crop often interferes with the results in real-time PCR and ELISA experiments [35]. Parallel reaction monitoring (PRM) analysis can also be performed on proteins associated with phenotypic traits to determine whether the inserted target gene has an effect at the proteome level, thus changing the plant’s phenotypic traits [36].

Rice insect pests have caused an annual worldwide grain loss of 10 million tons, which is equivalent to an economic loss of nearly 10 billion U.S. dollars [37]. In China, the main rice pests include rice planthoppers, stem borers, rice gall midges, and rice thrips. The cultivation of genetically modified rice is currently one of the main measures used to improve pest resistance [38]. China has developed multiple transgenic rice lines that expressed one or two of the several *CRY* toxins derived from *Bacillus thuringiensis* (Bt), of which *CRY1C* and *CRY2A* had shown better insecticidal effect against the *C. suppressalis* larvae [39]. In the present study, we compared the phenotypic and molecular variation as well as the level of protein expression in three transgenic rice restorer lines and their corresponding parents.

## 2. Results

### 2.1. Target Gene Insertion Site Analysis

We first developed MH63(1C), MH63(2A), and MH63(1C+2A) transgenic lines by inserting *CRY1C*, *CRY2A*, and both *CRY1C* and *CRY2A* genes, respectively. We then created three transgenic restorer lines [CH891(1C), CH891(2A), and CH891(1C+2A)] from BC_4_F_5_ generation by backcrossing MH63(1C), MH63(2A), and MH63(1C+2A), respectively, with a traditional rice variety CH891 (Figure 1). The *CRY1C* flanking of the insertion site sequence in CH891(1C) and CH891(1C+2A) was 1276 bp, of which 394 bp was derived from the T-DNA region. The *CRY2A* flanking sequence of the insertion site in CH891(2A) and CH891(1C+2A) was 948 bp, of which 356 bp was derived from the T-DNA region. *CRY1C* and *CRY2A* sequences were blasted on rice chromosomes 11 and 12, respectively. T-DNA was inserted in the 5′-noncoding region of a gene encoding an unknown functional protein (Figure 2).

### 2.2. Phenotypic Performance of the Transgenic Lines

As compared with the CH891 recurrent parent, three transgenic restorer lines [CH891(1C), CH891(2A), and CH891(1C+2A)] all displayed significantly greater resistance to insects at the tillering and heading stages, significantly smaller deadheart stems at the tillering stage, and significantly smaller whitehead panicles at the heading stage (Appendix A, Figure 3a and Appendix A). CH891(1C+2A) that had both the *CRY1C* and *CRY2A* genes gad the lowest insect damage. The three transgenic restorer lines and their recurrent parent were evaluated in one environment for 12 agronomic and end-use quality traits at paddy field conditions. Least significance difference (LSD) showed significant differences among the lines for seven traits, which included seed-set rate, brown rice rate, head rice rate, grain length to width ratio, chalkiness degree, and amylose content (Appendix A). Compared with the recurrent parent CH891, CH891(1C) showed significantly different brown rice rates, head rice rates, and chalkiness degree. CH891(2A) showed significantly different grain length-to-width ratios and amylose content. CH891(1C+2A) showed significantly different seed-set rates, brown rice rates, and amylose content (Figure 3b–l).

The gene expression levels analyses using qRT-PCR showed significantly greater *CRY1C* and *CRY2A* enrichment in all three transgenic restorer lines than their recurrent parent both at the tillering and heading stages; however, the level of both genes at the heading stage was 2–5-fold greater than at the tillering stage (Figure 4a, Appendix A). The Cry1C and Cry2A protein content measured by ELISA was basically the same in three transgenic lines at both tillering and heading stages. The Cry2A protein content was significantly greater than the Cry1C protein content measured by ELISA (Figure 4b, Appendix A). 

### 2.3. Molecular Variation

Of the 512 microsatellite markers used for genotyping the three transgenic restorer lines and their recurrent parent (CH891), 22 markers were polymorphic between CH891(1C) and CH891, accounting for 4.3% of total markers. There were 15 polymorphic markers between CH891(2A) and CH891 (2.9%), and 19 polymorphic markers between CH891(1C+2A) and CH891 (3.7%). We considered the 2.9–4.3% polymorphism observed in the transgenic lines as indicators of the donor genome content originated from Minghui 63 and the remaining (96.3–97.1%) as the recurrent parent genome originated from CH891, which agrees with the mean theoretical expectation of lines derived after four generations of backcrossing (Figure 5, Appendix A). In addition, the response rate of the actual genetic background of CH891(1C), CH891(2A), and CH891(1C+2A) selected was 97.85%, 98.54%, and 98.14%, respectively. The response rate of the theoretical genetic background of CH891(1C), CH891(2A), and CH891(1C+2A) selected was 96.875%, respectively. The actual genetic background response rate was higher than the theoretical genetic background response rate (Appendix A).

### 2.4. Proteome Analysis

We used the label-free quantitative proteomics method to compare the level of protein expression in the three transgenic lines and their recurrent parent from leaves sampled at the heading stage. The analysis performed on three biological replicates per line detected a total of 6650 proteins, of which 6324 were quantified. The diagonal line of the correlation cluster heatmap showed good repeatability of protein data during the overall analysis of the three transgenic lines and recurrent parents (Figure 6a). PCA also showed good repeatability (Appendix A). The mean protein quantitation value of CH891(2A) was 0.115, which was relatively smaller than the the0.202, 0.183 and 0.178 measured in the CH891, CH891(1C+2A), and CH891(1C), respectively (Figure 6b). We determine the number of differentially expressed proteins (DEPs) by comparing the transgenic lines with their recurrent parent (Table 1). The number of upregulated proteins in the transgenic lines varied from 168 in the CH891(2A) to 239 in the CH891(1C). The number of down-regulated proteins ranged from 131 in the CH891(2A) to 199 in the CH891(1C).

We next performed Clusters of Orthologous Groups (COGs) pathway enrichment analysis of DEPs. We mainly focused on COGs enriched by significant DEPs in the three groups (Figure 7, Appendix A). The number of DEPs = 214, 314, 319 represented the conserved protein under all of the COG pathways in CH891(1C), CH891(2A), and CH891(1C+2A), respectively (Appendix A). The COG pathways were mainly enriched in secondary metabolites biosynthesis transport and catabolism, carbohydrate transport and metabolism, post-translational modification, and signal transduction mechanisms. We also considered some unknown function pathways. The number of DEPs = 105, 154, and 150 represented the conserved protein under the five representative COG pathways in CH891(1C), CH891(2A), and CH891(1C+2A), respectively (Appendix A). We paid more attention to the top 5 up and 5 down-regulated proteins of five representative COG pathways among three transgenic and control lines, totaling 50 proteins. However, 26 proteins were screened from COG (the five representative pathways), which all existed among three transgenic lines (Table 2). Some proteins enriched in the COG pathways were also significantly enriched in the signal transduction mechanisms. B8AX75C and A2WKD1 were significantly upregulated, B8B2P3 and B8B281 were significantly downregulated in CH891(1C) and CH891(2A), A2XOW6(MAPK) in the signal transduction mechanisms were significantly upregulated in three transgenic lines in the pathway diagram (Figure 8). Some proteins enriched in the COG pathways were also significantly enriched in the KEGG pathways. The KEGG enrichment pathways mainly included glycine, serine, and threonine metabolism and tryptophan biosynthesis (Figure 9, Appendix A). Among the KEGG pathways significantly enriched in the upregulation and downregulation of DEPs were secondary metabolites biosynthesis transport and catabolism, and both showed enrichment of the three transgenic lines and control lines (Figure 10). Most DEPs in the glycine, serine, and threonine metabolism pathway were significantly upregulated in CH891(1C), CH891(2A), CH891(1C+2A), and CH891, and only a few DEPs were significantly downregulated in the pathway diagram (Figure 11). Most of these enzymes were downregulated. Information on these key DEPs includes enzyme symbol, EC number, and fold change. They included zinc finger matrin-type protein (ZMAT, 2.06, 1.59, 1.32), ubiquitin family protein (UBE, 4.12, 2.33, 4.18), EF-hand domain-containing protein (EFHC, 1.53, 1.56, 1.95), mitogen-activated protein kinase (MAPK, EC:2.7.11.25, 3.28, 2.40, 3.73), rhomboid-like protein (RHBD, 1.76, 2.11, 1.63), glutathione synthetase (GSS, EC:6.3.2.3, 2.45, 0.71, 2.75), C-factor (predicted) (CF, 3.69, 2.98, 1.47), and UDP-glucose 6-dehydrogenase (UG6D, EC:1.1.1.22, 3.61, 0.96, 4.99). Moreover, upregulated were enzymes of the cyanoamino acid metabolism pathway, such as UBX domain-containing protein (PUX, 0.64, 0.75, 0.91), protease Do-like 5, chloroplast precursor (PDI, 0.64, 0.66, 0.76), plant intracellular Ras-group-related LRR protein (PIRL, 0.62, 1.01, 0.60), zeta-carotene desaturase (ZDS, 0.63, 0.85, 0.87), tyrosine-protein phosphatase (PTP, EC:3.1.3.48, 0.49, 0.58, 0.60), serine hydroxymethyltransferase (SHMT, EC:2.1.2.1, 0.54, 0.59, 0.65), nicotinate phosphoribosyltransferase (NAPRT, EC2.4.2.11, 0.51, 0.69, 0.42), and hydroxy acid dehydrogenase (PGM, EC:5.4.2.2, 0.51, 0.99, 0.74). These DEPs were significantly enriched in some basic biologically conserved signal transduction mechanisms, KEGG pathways, and COG and belonged to constitutively expressed proteins (Figure 7, Table 2). Under the two insertion methods of *CRY1C* and *CRY2A genes*, the upregulation/downregulation trends of these DEPs were the same between the three transgenic lines and recurrent parents, and the changes at the proteome level were universal and representative and could be used for subsequent evaluation of unexpected effects.

### 2.5. PRM Validation

For proteome data validation, we selected genes encoding fifteen DEPs (MAPK, UBE, EFHC, SHMT, RHBD, GSS, CF, UG6D, PUX, PDI, PIRL, ZDS, PTP, NAPRT, and PGM) from the proteome data screened by cluster analysis. We performed PRM analysis on peptides representing these fifteen DEPs that were successfully quantified in the proteomic work (Table 3). Compared with the recurrent parents, ZMAT, UBE, EFHC, MAPK, RHBD, GSS, CF, and UG6D were significantly upregulated, and PUX, PDI, PIRL, ZDS, PTP, SHMT, NAPRT, and PGM were significantly lower in the three transgenic lines. On the other hand, except for ZDS, no significant difference was observed in DEPs among the three transgenic lines. The PRM results also correlated well with the proteomics data (Figure 12, Table 3). Therefore, the proteome data were reasonable and reliable.

## 3. Discussion

*CRY1Ab/1Ac*, *CRY1C*, and *CRY2A* are widely used Bt insect-resistance genes to enhance resistance to insect larvae without significantly affecting agronomic traits and end-use quality traits [8]. However, the transgenic lines may display unexpected phenotypes (linkage drag) due to the random *Agrobacterium*-mediated transformation, which often occurs in the transgenic donor parents prior to line conversion using backcrossing [17]. We tested whether transgenic technology in rice would have new or larger unexpected effects on conventional hybrid crops, using three transgenic lines of BC_4_F_3_ and one recurrent parent line.

In CH891 as the genetic background, *CRY1C* and *CRY2A* genes could be effectively and stably expressed in both transcription and translation stages to improve and maintain insect resistance. However, the expression level and resistance of different Bt proteins in the same genetic background are often different. Resistance of all Bt Bollgard cotton lines was inconsistent [40]. Cry1C and Cry2A proteins were also expressed differently in different rice lines. However, all transgenic lines carrying Cry1C or Cry2A exhibited nearly 100% resistance to C. suppressalis [7]. The higher expression of Cry1C and Cry2A proteins in leaves than in stems can be explained by spatial differences in the plant itself, as leaves are the more susceptible part of the plant [41]. The higher expression of Cry1C and Cry2A proteins at the heading stage than at the tillering stage can be explained by differences in physiological development in the plant itself because the heading stage was in the reproductive development stage with the most vigorous metabolism [39]. Therefore, in our study, *CRY1C* and *CRY2A* genes were stably expressed in high-generation backcross lines, and effective resistance to *C. suppressalis* was generated at different insertion sites.

Genomics, transcriptomics, and proteomics have been widely used to assess the effects of transgenic technology on crop breeding. The presence of a higher-than-expected donor genome in the backcross restorer lines has been cited as one of the factors that cause unintended phenotypic effects in transgenic crops [42,43,44]. Thus, more accurate 4D proteomics methods are needed to measure omics data [45]. Using 512 microsatellite markers distributed in all 12 rice chromosomes, we found out that the three transgenic lines inherited 2.9–4.3% of the donor parent (Minghui 63) genome, with the remaining 96.3–97.1% of the genome originating from the recurrent parent (Figure 4). These results agree with the theoretical expectations for lines developed after four generations of backcrossing breeding [9]. The comprehensive genetic background analysis showed that when the correlation within the group was high, differences between groups amplified in the PCA diagram, indicating the interference of a certain genetic background. This result was consistent with previous studies. There are some differences in protein components and characteristics among different transgenic lines [46,47].

The differential protein expression analyses revealed the presence of 168–239 upregulated genes and 131–199 downregulated genes in the transgenic lines than their recurrent parent (Table 1, Appendix A). The proteome expression levels among biological replicates of each line were highly correlated (0.93 ≤ r ≤ 0.99), which suggests the high reproducibility/consistency within each line. In contrast to our study, previous studies found no statistically significant difference in the transcriptome or proteome levels between transgenic and non-transgenic crops [48,49]. In beans, the similarity in leaf proteome between EMBRAPA and its non-transgenic near-isogenic line was higher than that between two common bean varieties [49]. To understand the relationship between phenotypic performance and gene expression, we evaluated the recurrent parent and the three transgenic lines for resistance against *C. suppressalis* larvae and the other 12 agronomic and end-use quality traits. As expected, the transgenic lines had significantly smaller larvae damage than their recurrent parent without affecting nine agronomic and end-use quality traits (Figure 3). The three other traits that showed statistically significant differences were brown rice rate, head rice rate, and grain length-to-width ratio, but there was no clear difference between the transgenic and non-transgenic lines as such. Such results suggest that the differentially expressed genes may not be directly involved in regulating the expression of the 12 agronomic and end-use quality traits in the transgenic lines. Alternatively, there may also be multiple other genes that we failed to detect that play major roles in regulating the agronomic and end-use quality traits. All the 12 phenotypic traits evaluated in the present study are quantitative in nature and can easily be affected by genotype-by-environment interaction. Such traits require replicated multi-environment experiments (at least three environments per trait), which was not the case in the present study.

We detected many DEPs between transgenic lines and recurrent parents. COG and KEGG enrichment analysis showed that DEPs in different comparisons were enriched in some constitutive metabolic pathways, mainly including carbohydrate transport and metabolism, coenzyme transport and metabolism, post-translational modification, and signal transduction mechanisms. Previous studies used four transgenic lines with insect resistance. The crystal proteins expressed in plants are heterologous, with no metabolic activity in rice plants [18]. However, if the inserted genes in transgenic plants were involved in plant metabolic pathways, they may cause plant phenotypic changes [50]. The results of this study were consistent with those of previous studies. Compared with traditional cross-breeding, transgenic plants have no unique influence on plant metabolic pathways.

PRM validation of 15 representative intra-group DEPs in the functional enrichment pathway found that most DEPs did not show significant differences among the three transgenic lines, indicating that the change in the proteome between transgenic lines would not lead to changes in grain yield and quality traits. Therefore, we attempted to analyze the relationship between polymorphic SSR markers and yield and quality traits to discuss the impact of the residue of the genetic background of donor parents on unexpected effects. *qPL9* was found to affect spike length in rice. An unsubstituted fragment of MH63 was found in CH891(1C), and *qPL9* was linked to the polymorphic SSR markers RM1026 [51]. *PTB1*, a ring E3 ubiquitin ligase, positively regulates the seed-setting rate of rice ears by promoting pollen tube growth. An unsubstituted fragment of MH63 was found in CH891(1C+2A), and *PTB1* was linked to the polymorphic SSR marker RM538 [52]. GW5 protein is a positive regulator of brassinolide signal transduction, which can physically interact with glycogen synthase kinase GSK2 and inhibit the activity of GSK2 kinase, regulate the expression level of brassinolide response gene, and affect grain width. An unsubstituted fragment of MH63 was found in CH891(2A), and *GW5* was linked to the polymorphic SSR marker RM289 [53]. *GL7* encodes a homolog of the LONGIFOLIA protein in Arabidopsis thaliana, which increases longitudinal cell division, decreases transverse cell division in grains and regulates longitudinal cell elongation. An unsubstituted fragment of MH63 was found in CH891(2A), and *GL7* was linked to the polymorphic SSR marker RM1364 [54]. *Wx* encodes granule-bound starch synthesis protein. An unsubstituted fragment of MH63 was found in CH891(1C+2A), and *Wx* was linked to the polymorphic SSR marker RM190 [55]. Yield traits (with easy phenotypic selection) were almost unchanged, whereas quality traits (with hard phenotypic observation) were changed more frequently, suggesting that the genetic background interfered with the evaluation of unintended effects of transgenes at the proteomic level in high-generation backcross lines. Our study showed that the internal difference in genetic background is much larger than the variation of plant proteome caused by the introduction of foreign genes by transgenic technology or cross-breeding. Due to the lack of environment tests, the inability to get a better understanding of the effects of genotype by environment interaction. The genetic stability and environmental adaptability of these transgenic lines will be further investigated in this study.

## 4. Materials and Methods

### 4.1. Plant Material and Phenotyping

Three transgenic Bt rice restorer lines, CH891(1C), CH891(2A), and CH891(1C+2A), and their non-Bt counterpart Changhui 891, were used in this study. Among these varieties, Changhui 891 is a backcross line. The seeds for these lines were independently cultivated at the Key Laboratory of Crop Physiology, Ecology, and Genetic Breeding, Ministry of Education, Jiangxi Agricultural University, Nanchang, China. Field experiments were performed in May–October 2021 in the economic and technological development zone (28°48′10″ N, 115°49′55″ E), Nanchang City, Jiangxi Province, China. The mean monthly day and night temperatures during the rice growing season are shown in Table 1. The three transgenic Bt rice lines and the control line, Changhui 891, were used for field evaluations. The field layout followed a randomized block design with three replications. The size of each plot was 5 m × 5 m. Twenty-day-old seedlings were transplanted at a density of 15 cm × 20 cm with one seedling per hill. The soil type of the experimental site was reddish-yellow clay-like paddy soil. The soil in the upper 15 cm at the test site had the following properties at the beginning of the experiment: pH 5.01, 1.26 g kg^−1^ total N, 105.6 mg kg^−1^ available phosphorus, 125.2 mg kg^−1^ potassium, and 20.56 g kg^−1^ organic matter. The experimental field was kept flooded from transplanting until seven days before maturity. Pests, diseases, and weeds were intensively controlled for all treatments to avoid yield losses.

In order to ensure the repeatability and accuracy of the experiment, three replicates were selected for individual plants at the heading stages, and 10 g of leaves were collected from three transgenic lines and a control line. Ten grams of the same tissue sample were collected for proteome and PRM; the excess was stored for future use. To exclude the interference of environmental factors, all samples were collected at 3 p.m. on the same day. Considering that proteins degrade easily, each sample was placed in a liquid nitrogen tank immediately after being harvested in the field and transported back to the laboratory, and stored at −80 °C.

Indoor insecticidal assays were conducted using ten replicates of each transgenic line and non-transgenic control. An artificial diet was administered to *C. suppressalis* larvae for 9–10 d, and most borers developed to the second instar stage by 10 d. Second instar larvae were placed on individual Petri dishes that contained a piece of the leaf (4 g) or stem (5 g), and ddH2O was added to the filter paper to keep the environment of the Petri dish humid. Petri dishes were sealed with parafilm membranes to prevent larvae from escaping. All Petri dishes were stored in a hermetic box in the dark at approximately 27 ± 1 °C and 70 ± 10% relative humidity. The insect resistance of transgenic plants in the field was assessed by artificially infesting rice plants with C. suppressalis. Chemical insecticides that target lepidopteran pests were not applied throughout the experimental period. Field assays were conducted using three replicates of each transgenic line and non-transgenic control. Thirty individual plants were planted in each replicate. At the tillering stage, 15–20 first-instar C. suppressalis larvae were applied to each rice plant. The number of dead hearts induced by stem borers was counted at the end of the maximum tillering stage, damaged leaves with visible scrapes or folds caused by leaf folders were counted within five days after peak damage appeared, and the number of white spikelets was counted at the flowering stage.

Transgenic rice lines were planted in paddy fields at the Transgenic Experimental Plots of Jiangxi Agricultural University (Nanchang, Jiangxi, China) to evaluate the agronomic performance. Non-transgenic control line CH891 was planted in paddy fields adjacent to the transgenic lines. Six blocks of 6 m^2^ (2 m × 3 m) were randomly chosen for the evaluation. Each block contained about 100 plants, and each plant had 10–15 tillers. Six yield traits and six quality traits were measured, including panicle length, panicles per plant, grains per panicle, weight for 1000-grain, seed set rate, and yield per plant. Filled and unfilled grains of the main panicle were separated manually for measurement of seed-setting rate (filled grains/(filled grains + unfilled grains) × 100). Yield per plant was calculated as panicles per plant × grains per panicle × Weight for 1000-grain × seed set rate × 10^−6^.

After maturity, all plants from each plot were harvested for measurements of brown rice rate, head rice rate, grain length to width ratio, chalkiness degree, gel consistency, and amylose content. The brown rice rate was calculated as the weight of brown rice grains divided by the weight of the entire sample of unhusked rice and taken as a percentage (using a sample of 300 g of total unhusked rice). Head rice rate was calculated as the weight of grains that were the same size or larger than 0.6 of the average length of the whole grain, measured with a rice grader, divided by the weight of the milled rice sample, and stated as a percentage (using a sample of 100 g milled rice). The grain length-to-width ratio was calculated as the grain length divided by the grain width. The chalkiness degree was calculated as the area of chalkiness rice divided by the total area of refined rice grains. Gel consistency was measured according to the flow characteristic of milled rice gel was measured in 0.2 N KOH and indexed by the length (in mm) of the cold horizontal gel (using a sample of 100 g 100-mesh-sieved rice flour). Amylose content was measured as involving a defatting step, where 95% ethanol was added to the milled rice flour prior to starch dispersion in NaOH, followed by gelatinization in a boiling water bath, and measurement of amylose based on the detection of iodine blue color at pH 4.5 to 4.8 using sodium acetate buffer (using a sample of 100 g 100-mesh sieved-rice flour) [56].

### 4.2. Detection of Target Gene Insertion Sites and Expression

DNA was extracted by the CTAB method. DNA was subjected to HindIII(Qut), and EcolI(Qut) restriction endonucleases in a 50-μL reaction system for 2 h, and the obtained fragments were cyclized in a T4 DNA ligase 50-μL reaction system. The obtained products were amplified by nested PCR, and primers were used for the first and second rounds of nested PCR (Appendix A). The first round of nested PCR included pre-denaturation at 94 °C for 5 min. The target sequences were enriched and recovered by 1% agarose gel electrophoresis and sent to Tsingke Biotechnology Company, Hunan Province, for sequencing.

Total RNA was extracted from tissues at various developmental stages by grinding in TRIzol. DNase digestion was performed to avoid contamination from genomic DNA, and the phenol–chloroform method was used to isolate total RNA. RNA quantity and quality were measured using a NanoDrop 2000 spectrophotometer (Thermo Fisher Scientific Inc., Waltham, MA, USA) based on the 260/280-nm and 260/230-nm absorbance ratios. Complementary DNA was synthesized using a PrimeScript 1st Strand cDNA Synthesis Kit (TaKaRa, 6210A, Kusatsu, Japan) according to the instructions of the PrimeScript RT Master Mix Kit. After ten-fold dilution of the cDNA, the target genes *CRY1C* and *CRY2A* and the reference gene Actin1 were detected by quantitative real-time PCR (qRT-PCR) according to the instructions of the SYBR Premix Taq II Kit (TaKaRa) (Appendix A). The purity of the amplicons was confirmed in the presence of a single peak in the melting curve [57]. Osactin1 was used as the internal reference gene for qRT-PCR normalization, and the qRT-PCR results were analyzed by the 2^−ΔΔCT^ method.

The amount of Cry1C and Cry2A proteins in leaves, stems, and panicles were measured at the tillering, booting, heading, filling, and maturity stages using an ELISA kit (AP003 and AP005 CRBS; EnviroLogix Inc., Portland, ME, USA). The absorbance value was measured at 450 nm using a VICTOR Nivo multimode plate reader (PerkinElmer, Waltham, MA, USA). Based on the range of the standard curve, the Cry1C and Cry2A protein extract was diluted appropriately so that their absorbance value was within the range of the standard curve [58]. For ELISA, a standard curve was drawn based on the absorbance of known concentrations of Cry1C and Cry2A standard (AP003 and AP005; EnviroLogix). The concentration of each test sample was determined from the standard curve, and the Cry1C and Cry2A protein content of the sample was calculated based on its dilution ratio and the conversion formula: Cry1C and Cry2A protein content (μg g^−1^ fresh weight) = test sample concentration (ng g^−1^) × dilution × extract volume/tissue fresh weight (mg) [34].

### 4.3. Genetic Background Detection Based on SSR Markers

A total of 512 SSR primers covering the whole genome of rice were used to screen the whole genome of the three transgenic lines and their corresponding recurrent parents, and the recovery rate of the genetic background was analyzed. The CTAB method was used to extract DNA. SSR primers were synthesized by Shanghai Shenggong Biotechnology Company based on the sequence designed by China Rice Research Institute. The PCR system was 15 μL, including 10 × PCR buffer 1.5 μL, 100 ng•μL^−1^ DNA template 2 μL, 2.5 mmol• L^−1^ dNTPs 0.3 μL, ddH_2_O 10 μL, 10 μmol• L^−1^ positive and negative primers 0.5, and 5U•μL^−1^ Taq DNA polymerase 0.2 μL. The PCR procedure was the same as above. The PCR products were observed after 8% polyacrylamide gel electrophoresis and rapid silver staining. The single plant of homozygous band type was marked as 1, the single plant of heterozygous band type was marked as 2, and the single plant of missing band type was marked as 0.

### 4.4. Protein Extraction and Liquid Chromatography (LC)-MS/MS Quantitative Proteomics

Leaf samples from three transgenic lines and recurrent parents at the heading stage were thoroughly ground to powder in liquid nitrogen at −80 °C. Phenol extraction buffer, four times the volume of the powder, was added to the samples and lysed by ultrasonication. 0.1 M ammonium acetate/methanol was added to the supernatant for overnight precipitation. The protein precipitate was washed with methanol and acetone. The protein concentration was determined using the BCA kit (Merck, B9643, Germany). Equal amounts of each sample protein were taken for enzymatic hydrolysis, and the volumes were adjusted to the same volume with lysis buffer. After drying the precipitate, the samples were subjected to in-gel tryptic digestion after reduction with DL-Dithiothreitol (DTT) at 60 qC for 30 min and alkylated with iodoacetamide (IAA) in the dark for 1 hr. Trypsin was added to the proteins at 1:50 and incubated at 37 °C overnight [59].

After separation by an ultra-high performance liquid phase system, the peptide was injected into the ion source for ionization using the Orbitrap Exploris™ 480 mass spectrometer (Thermo Fisher Scientific). The fixed start point of the secondary mass spectrum scanning range were all detected and analyzed using the high-resolution Top15 [60]. The secondary mass spectrometry data of this experiment were retrieved using Proteome Discoverer (V2.4.1.15). Retrieval parameter Settings: The database is Blast_Oryza_sativa_subsp._indica_39946_PR_20210107.fasta. According to the sequence of HCD-1.0 ion fragmentation, the secondary ion fragmentation mode was used for analysis. To improve the effective utilization of MS, the maximum implantation time was set to auto, and the dynamic exclusion time of tandem MS (MS/MS) scanning was set to 20 s to avoid repeated scanning of parent ions. For each sample, the quantification was normalized using the average ratio of all the unique peptides. Protein quantitation is calculated from the median ratio of protein corresponding to unique peptides. Three biological replicates were performed for each sample, and three technical replicates were performed for each biological replicate. Two-sided T-tests were performed to evaluate abundance changes of corresponding protein [61]. The MS proteomics data are available at the ProteomeXchange Consortium via the PRIDE partner repository (https://www.ebi.ac.uk/pride/, accessed on 28 April 2022) with the dataset identifier PXD033443.

### 4.5. PRM Analysis

Protein extraction was performed in line with the 4D label-free proteome method. The digested peptides were dissolved in 0.1% formic acid (solvent A) and directly loaded onto a customized reversed-phase analytical column. All at a constant flow rate on an EASY-nLC 1000 ultra-performance LC (UPLC) system (Thermo Fisher Scientific). The peptides were subjected to NSI source, followed by MS/MS in Q ExactiveTM Plus (Thermo Fisher Scientific) with online UPLC. A data-independent procedure that alternated between one MS scan followed by 20 MS/MS scans was followed. The resulting MS data were processed using Skyline (v.3.6). Peptide settings: enzyme was set as trypsin [KR/P], maximum missed cleavage was set as 2, peptide length was set as 8–25, the variable modification was set as carbamidomethyl on Cys and oxidation on Met, and maximum variable modification was set as 3. Transition settings: precursor charges were set at 2, 3, ion charges were set at 1, 2, and ion types were set at b, y, p. The product ions were set from ion 3 to the last ion; the ion match tolerance was set as 0.02 Da. After normalizing the quantitative information by the heavy isotope-labeled peptide, a relative quantitative analysis (three biological replicates) was performed on the target peptides [62].

### 4.6. Data Analysis

Vector NTI Suite 8 software (Invitrogen Corp., Carlsbad, CA, USA) was used to compare the flanking sequence with the transformed Vector PPZP201-rubisk-BT to determine whether the obtained sequence was the target fragment and remove the part identical to the Vector sequence. Sequencing results were analyzed for similarity using the NCBI database (http://last.ncbi.nlm.nih.gov, accessed on 12 February 2022). The GRAMENE library (http://www.gramene.org, accessed on 30 October 2022) was used to retrieve and analyze the rice genome sequence. After PCR band type sorting, data was inputted into a Microsoft Excel 2007 spreadsheet (Microsoft Corp., Redmond, WA, USA) for data processing, and CASS2.1 software was used to draw linkage maps. The regression rate of the recurrent parent background was calculated by the formula E[G(g)] = 1 − (½) ^g+1^. The common formula G(g) = [L + X(g)]/2L was used to calculate the response rate of the genetic background in actual analysis, where G(g) represented the response rate of the genetic background in g generation. g represents the number of generations used for backcrossing; L represents the number of molecular markers involved in analysis; X(g) represents the number of band markers in the backcross g generation, which were the same as those of recurrent parents.

The protein data with quantitative values in all samples were selected for dimensionality reduction. Data were first transformed by log_2_, and the mean value was subtracted. Then the PRCOMP function in R was used for principal component analysis (PCA). Protein annotation mainly included the Kyoto Encyclopedia of Genes and Genomes (KEGG) annotation and subcellular localization. The KEGG database was used to annotate the protein pathways. First, BLAST comparison (blastp, evalue ≤ 1e^−4^) was performed on the identified proteins. Based on the BLAST comparison results, sequences with the highest scores were selected for annotation. Hierarchical clustering was performed based on differentially expressed protein functional classification (KEGG Pathway). All enriched categories, along with their *p*-values, were first collated, then those categories enriched in at least one of the clusters with *p* < 0.05 were filtered. This filtered *p*-value matrix was transformed using the function x = −log_10_ (*p*-value) [63]. KEGG online service tools KAAS (https://www.genome.jp/tools/kaas/, accessed on 30 October 2022) was used to annotate the KEGG database description of proteins [64]. Finally, the annotation results of the KEGG database pathways were mapped using KEGG online service tool KEGG mapper. These pathways were classified into hierarchical categories using the KEGG website. In the experimental design, more than two unique peptides were used for the quantitative analysis of each protein (only one unique peptide was suitable for PRM verification for some proteins), and only one peptide was identified for some proteins due to sensitivity and other reasons. After normalizing the quantitative information by the heavy isotope-labeled peptide, a relative quantitative analysis (three biological replicates) was performed on the target proteins. The protein relative expression levels were processed and analyzed using Excel 2007 and SPSS 16.00 (IBM Corp., Armonk, NY, USA).

Agronomic traits of transgenic plants were compared with the recurrent parents using one-way analysis of variance (ANOVA). Values were presented as means (±SD). The borer mortality rate data were processed and analyzed using Excel 2007 and SPSS 16.00. Figures were constructed using Origin 2017 (OriginLab Corp., Northampton, MA, USA).

## 5. Conclusions

We successfully used 4D label-free quantitative proteomics technology to investigate the changes in protein expression by functional clustering in three transgenic lines and recurrent parents. The results show that the inserted *CRY1C* and *CRY2A* genes were inherited stably in higher generations, and the newly bred transgenic restorer lines showed high insect resistance and superior agronomic traits, which can lead to changes in the proteome. Moreover, no negative or unintended effects of proteomic changes were observed on grain yield and quality traits of these transgenic lines. Thus, this new omics technology can provide an effective detection method for identifying the unintended effects of transgenic varieties. We also found that the residual fragments of donor genetic background during backcross selection may have more influence on the agronomic traits of transgenic varieties. Therefore, it is necessary to establish a comprehensive evaluation system of unintended effects at the multi-omics analysis level of commercial variety selection before the analysis of unintended effects is considered in this study.

## Figures and Tables

**Figure 1 plants-12-00156-f001:**
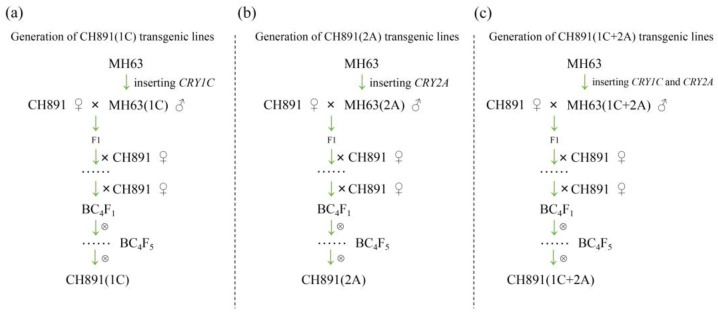
The development of transgenic restorer lines: (**a**) CH891(1C) was produced by backcrossing MH63(1C) as a donor parent with CH891 as a recurrent parent; (**b**) CH891(2A) was produced by backcrossing MH63(2A) with CH891, and (**c**) CH891(1C+2A) was produced by backcrossing MH63(1C+2A) with CH891.

**Figure 2 plants-12-00156-f002:**
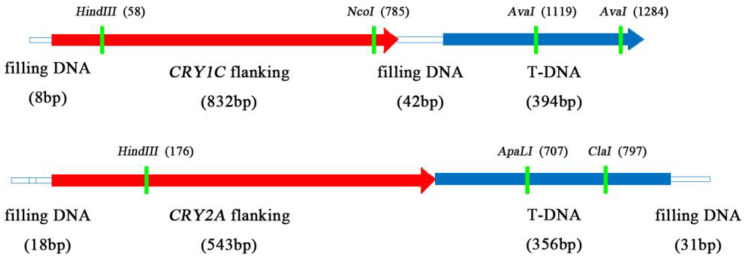
Schematic map of flanking sequences of *CRY1C* and *CRY2A* insertion sites.

**Figure 3 plants-12-00156-f003:**
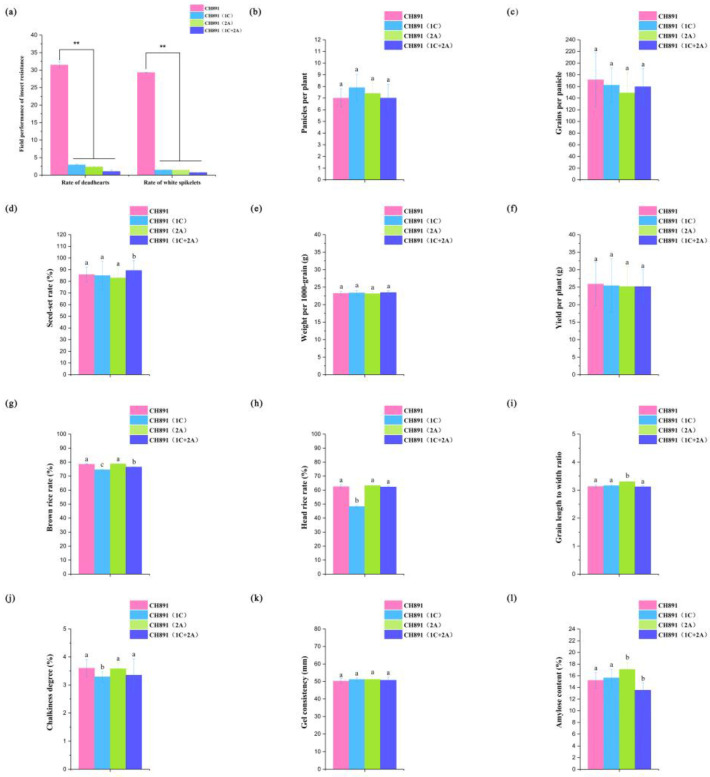
Comparison of a recurrent parent (CH891) with CH891(1C), CH891(2A), and CH891(1C+2A) transgenic restorer lines that resistance to *C. suppressalis*, and agronomic traits, respectively. (**a**). The reaction of the three transgenic restorer lines and their parent on insect larvae damage under natural infestation at the tillering and heading stage in Nanchang field (E115.5°, N28.5°), China. (**b**–**l**). Bar charts of the 12 agronomic and end-use quality traits recorded in three transgenic lines and their recurrent parent. The phenotype data were based on one environment. The bars indicate the standard deviations, and the median horizontal line indicates the mean. See Appendix A for Pairwise comparisons. Data of each test were analyzed by one-way ANOVA (** *p* < 0.01). Data followed by different lower-case letters denote significant differences between agronomic traits at the 5% level according to LSD test.

**Figure 4 plants-12-00156-f004:**
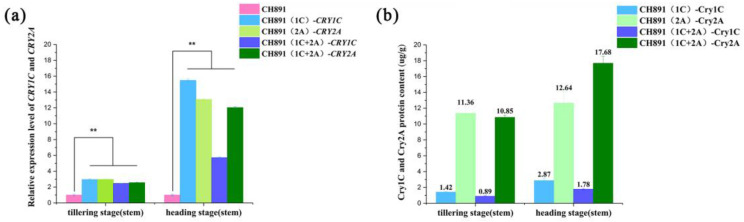
Comparison of a recurrent parent (CH891) with CH891(1C), CH891(2A), and CH891(1C+2A) transgenic restorer lines that consisted of *CRY1C, CRY2A*, and both *CRY1C*/*CRY2A* genes, respectively. (**a**). Relative gene expression levels of *CRY1C* and *CRY2A* based on qRT-PCR. (**b**). Cry1C and Cry2A protein content was detected by enzyme-linked immunosorbent assay (ELISA). Data of each test were analyzed by one-way ANOVA (** *p* < 0.01).

**Figure 5 plants-12-00156-f005:**
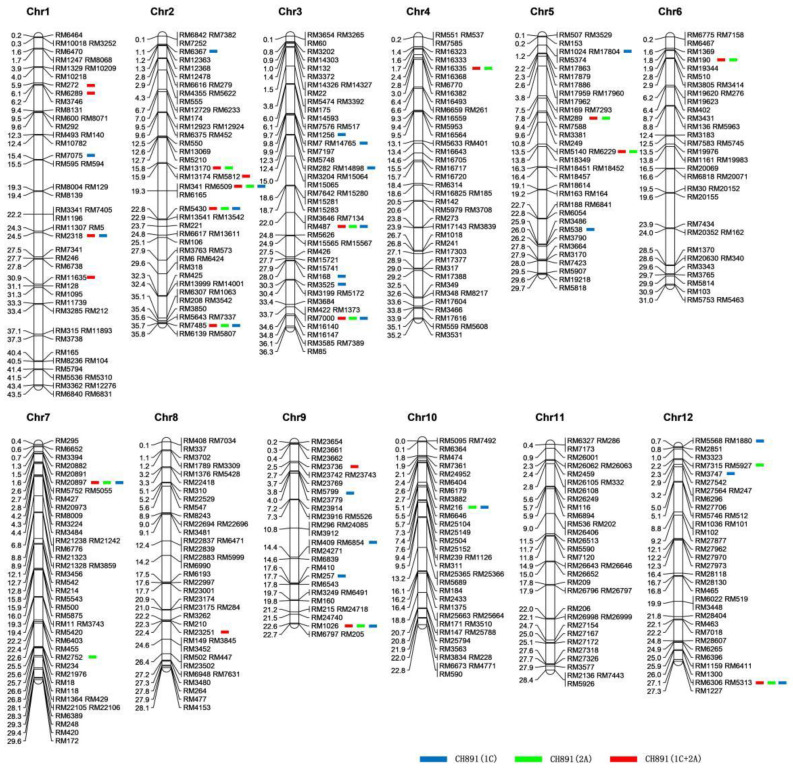
The chromosomal distributions of the 512 microsatellite markers used for genotyping CH891(1C), CH891(2A), CH891(1C+2A), and CH891. Blue, green, and red bars represent the homozygous markers for Minghui 63 genome and/or the Bt genes in three restored lines [CH891(1C), CH891(2A), and CH891(1C+2A)], respectively; the remaining markers were homozygous for the CH891 genome. The scale on the left indicates the physical positions (Mb) of each marker, while the right side of the map shows the name of the microsatellite markers.

**Figure 6 plants-12-00156-f006:**
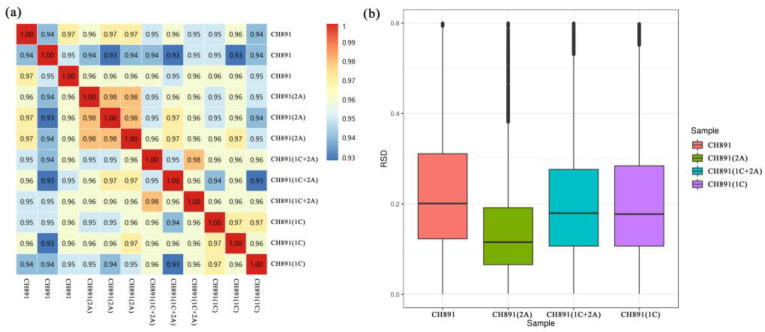
(**a**) Correlation coefficients and (**b**) box plots of protein expression levels of leaf samples of three transgenic lines and their recurrent parent collected at the heading stage. Each line was represented by three biological replicates. RSD (relative standard deviation), boxplot of RSD of quantitative protein values between repeated samples. The smaller the overall RSD value, the better the quantitative repeatability.

**Figure 7 plants-12-00156-f007:**
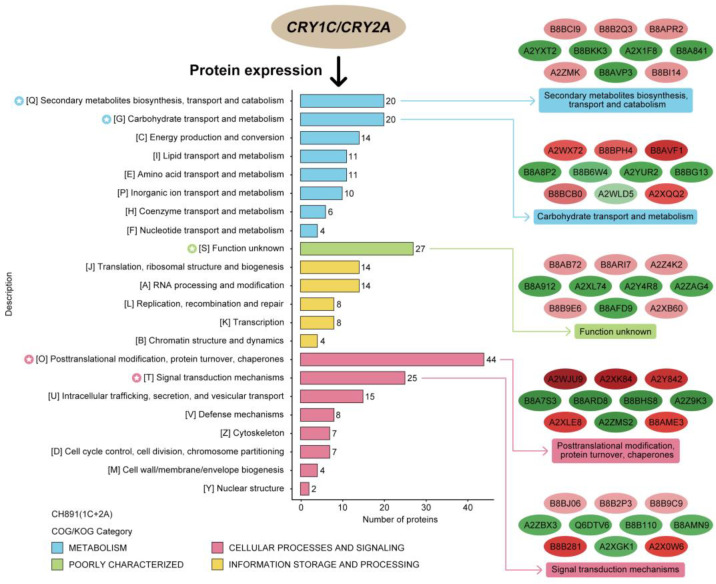
Cluster analysis of differentially expressed proteins (DEPs). Clusters of Orthologous Groups of proteins (COG) displayed the degree of accumulation of conserved proteins among samples of NIL groups CH891(1C+2A).

**Figure 8 plants-12-00156-f008:**
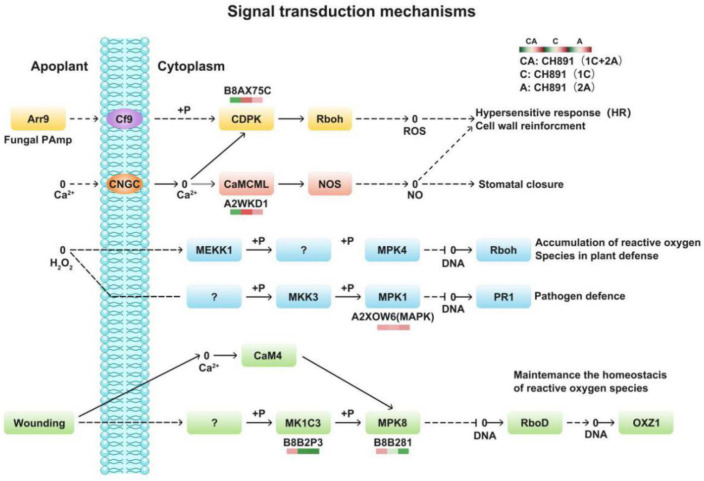
Changes in quantified proteins associated with the signal transduction mechanisms in three transgenic lines (NIL groups) and control lines (left box: CH891(1C+2A)/CH891, center box: CH891(1C)/CH891, right box: CH891(2A)/CH891).

**Figure 9 plants-12-00156-f009:**
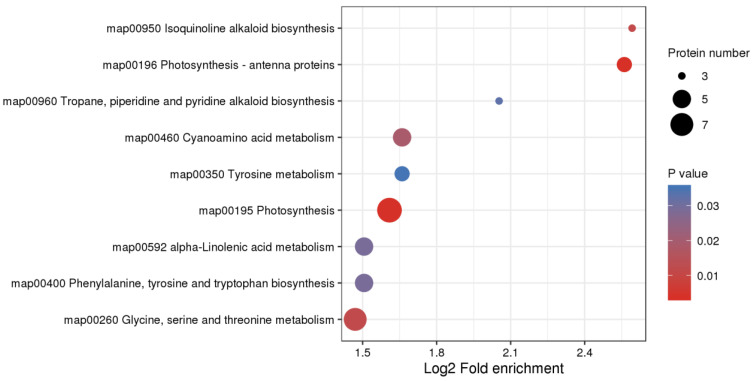
KEGG enrichment analysis of differentially expressed proteins (DEPs). KEGG enrichment analysis was performed for each protein class by Fisher’s exact test method, and the enrichment results for different classifications were combined. Then, the top 40 (*p* < 0.05) significantly enriched functional classifications were identified. The horizontal axis shows fold enrichment after Log_2_ transformation, the vertical axis shows the functional classification, the bubble size represents the number of proteins, and the bubble color represents the value of *p* of enrichment significance.

**Figure 10 plants-12-00156-f010:**
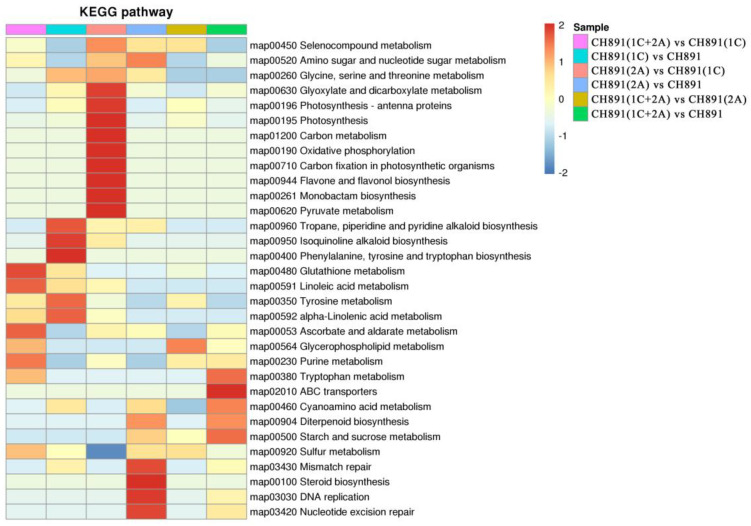
Kyoto Encyclopedia of Genes and Genomes pathways significantly enriched in sets of DEPs among samples of NIL groups. The enriched pathways included metabolic pathways by which secondary metabolites biosynthesis transport and catabolism.

**Figure 11 plants-12-00156-f011:**
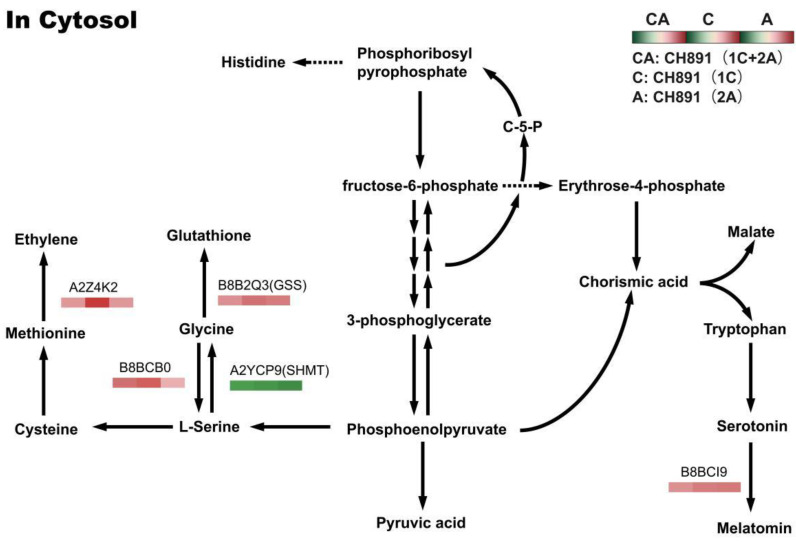
Changes in quantified proteins associated with the glycine, serine, and threonine metabolism pathways in three transgenic lines (NIL groups) and control lines (left box: CH891(1C+2A)/CH891, center box: CH891(1C)/CH891, right box: CH891(2A)/CH891).

**Figure 12 plants-12-00156-f012:**
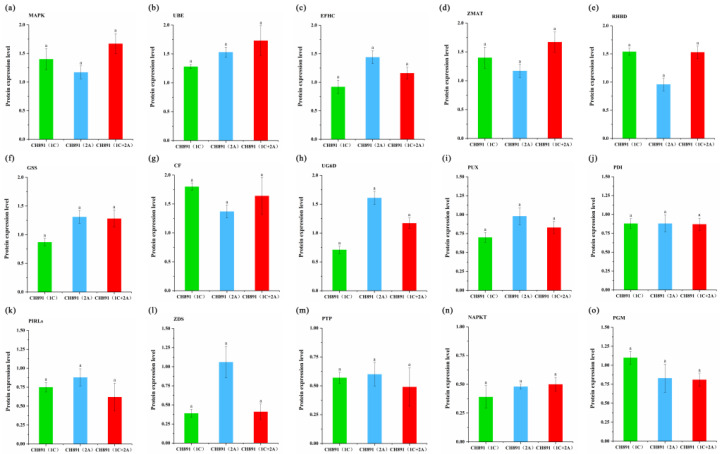
Parallel reaction monitoring (PRM) of proteins involved in the COG enrichment pathways by three transgenic events. Samples were obtained from leaves at the heading stage in CH891(1C), CH891(2A), and CH891(1C+2A). (**a**–**o**) represents PRM results of key enzymes MAPK, UBE, EFHC, SHMT, PPI, GSS, CF, UG6D, PUX, PDI, PIRLs, ZDS, PTP, NAPKT, and PGM, respectively. Relative protein rates of each sample were compared to those in CH891. The data are means of three biological replicates; different letters indicate significant differences within NIL groups (*p* < 0.05), and bars between columns indicate significant differences among proteins among NIL groups. Data followed by different lower-case letters denote significant differences between relative protein rates of each sample at the 5% level according to LSD test.

**Table 1 plants-12-00156-t001:** Summary of the number of differentially expressed proteins (DEPs) in three transgenic lines [CH891(1C), CH891(2A), and CH891(1C+2A)] as compared with their recurrent parent (CH891). The number of DEPs was determined using a threshold ratio of 1.5-fold between those proteins expressed in each transgenic line to those expressed in the recurrent parent. Upregulated proteins had a ratio of >1.5-fold and downregulated proteins had a ratio of <1/1.5-fold.

Compared Sample Name	Up-Regulated Protein	Down-Regulated Protein
CH891(1C) vs. CH891	239	199
CH891(2A) vs. CH891	168	131
CH891(1C+2A) vs. CH891	198	198
[CH891(1C) vs. CH891] ∩ [CH891(2A) vs. CH891]	41	33
[CH891(1C) vs. CH891] ∩ [CH891(1C+2A) vs. CH891]	39	48
[CH891(2A) vs. CH891] ∩ [CH891(1C+2A) vs. CH891]	52	54
[CH891(1C) vs. CH891] ∩ [CH891(2A) vs. CH891] ∩ [CH891(1C+2A) vs. CH891]	22	23

**Table 2 plants-12-00156-t002:** Functional classification of key differential proteins in COGs.

Protein Symbol	Protein Description	CH891(1C)/CH891	CH891(2A)/CH891	CH891(1C+2A)/CH891
B8A8P2	1,4-alpha-D-glucan glucanohydrolase	0.526	0.758	0.822
A2YUR2	Tyrosine-protein phosphatase domain-containing protein	0.493	0.581	0.603
A2YCP9	Serine hydroxymethyltransferase	0.543	0.593	0.654
B8AN97	Nicotinate phosphoribosyltransferase	0.511	0.685	0.422
B8AC53	MoCF_biosynth domain-containing protein	0.591	0.665	0.659
B8BHS8	J domain-containing protein	0.490	1.200	0.593
A2ZMS2	Protease Do-like 5, chloroplast, putative, expressed	0.643	0.656	0.748
A2ZAG4	Plant intracellular Ras-group-related LRR protein 5	0.624	1.011	0.599
A2ZBX3	Calcium-dependent protein kinase 24	0.567	0.574	0.812
B8BPH4	UDP-glucose 6-dehydrogenase	3.608	0.961	4.989
B8AVF1	OSIGBa0106G07.1 protein	7.107	5.766	4.810
A2WJU9	Peptidyl-prolyl cis-trans isomerase	1.850	2.073	1.081
A2XLE8	Matrin-type domain-containing protein	2.063	1.591	1.319
B8AME3	Ubiquitin family protein, expressed	4.128	2.330	4.183
B8BCI9	Fe_2_OG dioxygenase domain-containing protein	3.572	2.699	2.746
B8B2Q3	Glutathione synthetase	2.445	0.709	2.754
B8APR2	Putative alcohol dehydrogenase	6.968	6.381	0.926
A2ZMK7	C-factor	3.691	2.980	1.467
B8B9E6	WD_REPEATS_REGION domain-containing protein	2.255	1.095	1.955
A2XB60	Acyl-CoA binding protein-like	1.705	1.482	1.832
B8BJ06	EF-hand domain-containing protein	1.532	1.555	1.951
B8B9C9	RHOMBOID-like protein	1.756	2.110	1.629
A2X0W6	Mitogen-activated protein kinase	3.280	2.389	3.729
B8B894	Zeta-carotene desaturase	0.630	0.851	0.866
B8BG13	Phosphoglucomutase	0.510	0.990	0.741
B8ARD8	UBX domain-containing protein	0.638	0.750	0.907

Note: The relative quantitative value of each protein in the two comparison samples was tested by *t*-test, and the corresponding *p*-value was calculated as the significance index, and the default *p*-value was ≤0.05. In order to make the test data conform to the normal distribution required by the *t*-test. Before the test, the relative quantification value of the protein should be Log_2_ log-transformed.

**Table 3 plants-12-00156-t003:** Peptide sequences of fifteen candidate proteins in PRM.

Protein Symbol	Peptide Sequence	Retention Time	CH891(1C)/CH891 Ratio	CH891(2A)/CH891 Ratio	CH891(1C+2A)/CH891 Ratio
A2XLE8 (ZMAT)	CEICGNHSYWGR	12.05	1.4	1.17	1.67
B8AME3 (UBE)	ALIATAGNVHAAVER	13.37	1.28	1.53	1.73
B8BJ06 (EFHC)	AIEYDNFIECCLTVK	23.95	0.92	1.44	1.16
A2X0W6 (MAPK)	YLHSAEILHR	10.5	0.91	1.36	0.99
B8B9C9 (RHBD)	SNAIEHAHFR	7.72	1.54	0.96	1.53
B8B2Q3 (GSS)	ELAPIFNDLVDR	25.83	0.87	1.31	1.28
A2ZMK7 (CF)	TALNQLTK	11.64	1.8	1.37	1.64
B8BPH4 (UG6D)	ETPAIDVCHGLLGDK	18.25	0.71	1.61	1.17
B8ARD8 (PUX)	AFHFVQPIPR	17.24	0.7	0.98	0.83
A2ZMS2 (PDI)	LVGCDPSYDLAVLK	21.92	0.88	0.88	0.87
A2ZAG4 (PIRLs)	VFDDLIQR	18.3	0.75	0.88	0.62
B8B894 (ZDS)	ALVDPDGALQQVR	18.22	0.39	1.06	0.41
A2YUR2 (PTP)	FIAGGQWR	15.28	0.57	0.6	0.49
B8AN97 (NAPRT)	AYVVPQHVEELLK	19.49	0.39	0.48	0.5
B8BG13 (PGM)	EHWATYGR	9.44	1.1	0.83	0.81

## Data Availability

The original contributions presented in the study are publicly available. The mass spectrometry proteomics data are available at the ProteomeXchange Consortium via the PRIDE partner repository with the dataset identifier PXD033443. The data sets supporting the results of this article are included within the article and its additional files. All experiment of plant and all field experiments are performed in your affiliated university, and the permissions of MS proteomics data are obtained at the ProteomeXchange Consortium via the PRIDE partner repository (https://www.ebi.ac.uk/pride/, accessed on 28 April 2022) with the dataset identifier PXD033443.

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
