# Peer review of "Comparison of the Phenotypic Performance, Molecular Diversity, and Proteomics in Transgenic Rice"

_plants, 2022, doi:10.3390/plants12010156_

Round 1
Reviewer 1 Report
Please see the attached 3 pages of comments.

Author Response
Dear Editors:
No.: 2035990
“Unintended effects of transgenic rice on grain yield and quality traits determined by quantitative proteomics” Thank you very much for your effort in handling our manuscript. We greatly appreciate the thoughtful and constructive comments from the reviewers and have revised the paper accordingly. In addition, we also added the latest author ranking and correspondence unit information in the manuscript. We use blue text in the document to highlight changes to the manuscript to help you see them better. Thank you for offering us the opportunity of revising the manuscript. Enclosed please find the detailed, itemized list of our responses to the comments of the reviewers and the changes we have made to the paper. With the incorporation of suggestions from reviewer, the manuscript has been obviously improved. We hope that the revised manuscript will satisfy the reviewers and that you will find the paper is now suitable for publication in Plants.
Thanks again for your timely consideration of our manuscript.
On behalf of the authors,
Xiaosong Peng
Comments to author:
Title
1、The manuscript tried to compare three transgenic lines with their recurrent parent using genotype data, protein analyses, and 12 phenotypic traits. The title of the manuscript is specific to protein analysis, which does not cover all methods used.
Response and Revision: Thank you very much for your meticulous comments and kind suggestions. Your advice has greatly benefited us. To make the title of the manuscript more accurate, we modify it to "Unintended effects of transgenic rice on grain yield and quality traits determined by genotype analysis and quantitative proteomics ". The 12 phenotypic traits represented 6 grain yield traits and 6 quality traits, respectively.
2、Lines 8-10 are two general and can be replaced by a stronger justification for your work. Did you conduct this study due to a lack of similar previous studies' or there were other studies that had weaknesses in methodology that can be improved using modern tools?
Response and Revision: Thank you very much for your meticulous comments and kind suggestions. Your advice has greatly benefited us. We conduct this study due to there were other studies that had weaknesses in methodology that can be improved using quantitative proteomics. The lines 8-10 is revised as “Previous unexpected effects on transgenic plants mainly focused on whether the insertion of target genes would affect the expression of genes and proteins near the insertion sites. However, such results can hardly tell whether there are differences at the whole genome and proteome level and whether they represent safety problems and their value for risk assessment is thus questionable”.
3、Unintended effects in backcross breeding are called linkage drag, which simply refers to a reduction in the performance of the line or cultivar due to deleterious genes introduced along with the beneficial gene https://link.springer.com/article/10.1007/s11032-013-9936-7; https://onlinelibrary.wiley.com/doi/full/10.1111/pbi.13576
Response and Revision: Thank you very much for your meticulous comments and kind suggestions. Your advice has greatly benefited us. We regulated the rigor of wording. We replaced "The unintended effect of transgenic organisms refers to the change in the original genetic traits caused by the insertion of foreign genes" with" Unintended effects in backcross breeding are called linkage drag, which simply refers to a reduction in the performance of the line or cultivar due to deleterious genes introduced along with the beneficial gene", and cited references [15] https://link.springer.com/article/10.1007/s11032-013-9936-7.
Abstract
4、I suggest revising the result section based on their main findings, which are clearer. If I understood correctly, the three transgenic lines had significantly lower stem borer infestation than the recurrent parent without showing significant differences among most of the agronomic traits, yield components, and end-use quality traits.
Response and Revision: Thank you very much for your meticulous comments and kind suggestions. Your advice has greatly benefited us. We replaced " which significantly improved the resistance of restorer lines to Chilo suppressalis" with" which the three transgenic lines had significantly lower stem borer infestation than the recurrent parent without showing significant differences among most of the agronomic traits, yield components, and end-use quality traits".
Introduction
5、The introduction section has multiple fragmented sentences that are hard to follow (there is no clear structure). This is especially true for lines 72-87 that is hard to understand. Focus on proteomics and delete the other methods. Briefly, provide a list of proteomics methods with citations followed by the methods that you used in this paper and a justification of how you have chosen them. In the results and methods sections, you have described qPCR, ELISA, LC-MS/MS, background analyses, COGS, DEPS, etc. without telling readers the purpose of each of these methods.
Response and Revision: Thank you very much for your meticulous comments and kind suggestions. Your advice has greatly benefited us. We removed everything else not related to proteomics. In the introduction section, the application methods of proteomics in previous studies and the reasons for choosing them are added, and the corresponding references of qPCR, ELISA, LC-MS/MS, background analysis, COGS, DEPS and other methods are added.
6、In lines 88-95, there is no description and citation for CRY1C and CRY2A, why these two genes were of interest in China, and if there were previous studies. The same is true for the C.suppressalis: what is this, how important is it in China, and what is the estimated loss due to this pest? These points should be reviewed in one paragraph to justify your work. Reviewing previous work and gaps that you are trying to address in this study.
Response and Revision: Thank you very much for your meticulous comments and kind suggestions. Your advice has greatly benefited us. We added the statement “Insect pests reduce rice production: the annual worldwide economic loss caused by rice pests amounts to nearly 10 billion U.S. dollars, and the annual loss of rice due to rice pests is close to 10 million tons [34]. In China, the main rice pests include rice planthoppers, stem borers, rice gall midges, and rice thrips. The cultivation of genetically modified rice is currently one of the main measures used to improve pest resistance [35]. In order to prevent suppressalis resistance, more Bt proteins were modified based on CRY1Ab, among which CRY1C and CRY2A had better insecticidal effect [36]”.
Results
7、All figures are very poor quality and hard to read. Image and font sizes should be higher, resolutions should be improved, and some irrelevant figures should be moved to the supplementary files. Some of the figures should be merged/combined (see examples below).
Response and Revision: Thank you very much for your meticulous comments and kind suggestions. Your advice has greatly benefited us. Our images are all in TIFF format, but are automatically reduced in sharpness by being inserted into the word version of the manuscript. We uploaded each figure in high resolution TIFF format. In addition, we included some pictures in the supplementary files according to your comments.
8、Lines 98-111 and Figure 1: The development of transgenic restorer lines is confusing. If I understood it, MH63 was transformed by inserting CRY1C and CRY2A genes, which resulted in three transgenes (with only CRY1C, only CRY2A, and both genes). Each of these transgenic lines was then crossed with CH891 to make F1. F1 plants were subsequently backcrossed with CH891 as a recurrent parent to BC4F5 generation, followed by the selection of CH891(1C), CH891(2A), and CH891(1C+2A) to represent transgenic restorer lines. If this is correct, Figure 1 should be modified to show all these steps and the nontransgenic control should be replaced by CH891.
Response and Revision: Thank you very much for your meticulous comments and kind suggestions. Your understanding is completely correct, we revised the Figure 1.
9、Lines 119 and 174: If the 3 trangenic lines were developed from CH891, there should be only one recurrent parent (but not two).
Response and Revision: Thank you very much for your meticulous comments and kind suggestions. Your advice has greatly benefited us. In order to make the expression more accurate and clear, we modified in Figure legend 1. 3 trangenic lines were only one recurrent parent. “Genetic relations of transgenic restorer lines in this study. a. Transgenic CH891(1C) was produced by backcrossing imports with the transgenic donor parent MH63(1C). b. Transgenic CH891(2A) was produced by backcrossing imports with the transgenic donor parent MH63(2A). c. Transgenic CH891(1C+2A) was produced by backcrossing imports with the transgenic donor parent MH63(1C+2A). Their recurrent parent is CH891”
10、Lines 117-135: The results summarized expression levels of three transgenic lines and their parents using two molecular methods (qPCR and ELISA) and field performance (insect resistance) at different stages (tillering, heading, panicle, stem, etc.), which are hard to follow. In my opinion, (i) you should compare the transgenic and nontransgenic plants using qPCR (one figure), followed by ELISA (another figure), and reaction to the pest (another figure). The reader should be able to compare differences among and within the transgenic and non-trangenic lines in the same figure or table (present figures b-d together and e-g together in the same ways as Figure i); (ii) both figures h and j do not show anything (I only see a seedling with no clear difference) and should be moved to Supplementary figure; (iii) figure i and k should be presented together (just modify the y-axis title and use a legend to show the deadhearts and white spikelets). I would start by telling readers whether there were statistically significant differences among the four lines for their reaction to the insect and explain if those differences are explained by the qPCR and ELISA.
Response and Revision: Thank you very much for your meticulous comments and kind suggestions. Your advice has greatly benefited us. We regulated the rigor of wording. We rearranged the logical order. First, the four lines for their reaction to the insect was significantly analyzed, and then qPCR and ELISA were used to explain the effect of CRY1C and CRY2A gene expression on insect resistance of rice. In addition, Figures 2b-2d was merged into one figure. Figures 2e-2g was merged into one figure. Figures 2e-2g is merged into one figure. Figure 2i and 2k was presented together. We moved both figures h and j to Supplementary Figure S1.
11、Lines 147-165 should be graphical genotyping to compare differences between each transgenic line and the parents. In the methods section, you only genotyped the 3 trangenic lines and parent but not a BC4F5 population. For that reason, Figure 3 should be replaced by a graphical genotype (not linkage maps) to show the region originating from MH63, CH891, the inserted genes, and heterozygous ones. See Figure 1 here as an example https://www.nature.com/articles/srep37505
Response and Revision: Thank you very much for your meticulous comments and kind suggestions. Your advice has greatly benefited us. Our three transgenic lines were all selected from BC4F5. Since our SSR was the initial public primers provided by Chinese rice for mapping cloning, it had clear linkage distance and high chromosome coverage. Through comparison with the SSR marker polymorphism of the parents, The genetic background response rate of the three transgenic lines can be calculated well.
12、The PCA in Figure 4 is not showing anything useful and suggests removing it. According to Figure 4c, the mean protein quantitative value for CH891(2A) is smaller than CH891, CH891(1C), and CH891(1C+2A) but the differences were not statistically significant, which are evident from the high correlation coefficients between pairs of the four lines in the heatmap. In my opinion, these results should be clearly stated in the text without making vague statements. You should also clearly describe why you have three samples for each line in Figure 4 (are they representing 3 replicates, sampling done at different stages, ..?)
Response and Revision: Thank you very much for your meticulous comments and kind suggestions. Your advice has greatly benefited us. Correlation and RSD analysis redescribed, which was modified to “Each plant line was measured by 3 biological replicates (Figure 4a). The mean protein quantitation values of CH891, CH891(2A), CH891(1C+2A)and CH891(1C) were 0.202, 0.115, 0.183 and 0.178, respectively. The relative standard deviation of protein quantitative values for CH891(2A) is smaller than CH891, CH891(1C), and CH891(1C+2A) but the differences were not statistically significant, which was consistent with the correlation analysis (Figure 4b).”. In addition, We moved Figure 5 to Supplementary Figure S2.
14、I suggest presenting Figure 5 in Table. This figure is not showing anything useful information, which can better be summarized in a table. Summarize only the number of genes that were up or down-regulated in each line.
Response and Revision: Thank you very much for your meticulous comments and kind suggestions. Your advice has greatly benefited us. The number of genes that were up or down-regulated were summarized in a table. We moved Figure 5 to Supplementary Figure S3. We added DEPs in Table 1.
15、Lines 302-319: This section compares the three transgenic lines with their parent across 12 phenotypic traits and is summarized in Figure 12. The authors presented both error bars and a test for significance, which are contradictory. You should either use the error bar in the figure or do ANOVA or mixed linear model analyses and present a table with F stat and p-values.
According to Figure 12 and the error bars, there were significant differences among the four lines only for brown rice yield, head rice rate, and grain length/width ratio. That should be clearly stated in this section. Alternatively, revise the results (text) supported by F stat and p values (in this case, you should delete Figure 12). I also strongly suggest moving the phenotypic performance results presented in lines 117-134 and Figure 2 here and include something like this: As compared with the recurrent parent, the three transgenic lines had significantly lower stem borer infestation without showing significant differences in most of the agronomic traits, yield components, and end-use quality traits. This is my understanding of the results.
Response and Revision: Thank you very much for your meticulous comments and kind suggestions. Your advice has greatly benefited us. We removed the significance test from the Figure 12 and explained the reasons for the significant differences in brown rice yield, head rice rate, and grain length/width ratio in the discussion section. Since the logic of this paper is to first determine the expected insect resistance effect of the transgenic lines, then analyze the changes of its proteome, and finally comprehensively analyze its influence on the yield and quality traits, we put the phenotypic data in the last paragraph of the results. We are very much in favor of your proposal. We very much agree with your viewpoint, which was added to the paragraph of results. As compared with the recurrent parent, the three transgenic lines had significantly lower stem borer infestation without showing significant differences in most of the agronomic traits, yield components, and end-use quality traits.
16、Lines 319-330: This section plus Table 3 compares donor chromosome fragments against the phenotypic traits but the results are very misleading. I don’t think you can simply link a known gene with a marker without providing the physical position. In my opinion, you can only compare the parent and transgenic lines at each polymorphic marker if they have different genotypes (something like Figure 3 in this link https://www.mdpi.com/2073-4395/12/7/1706/htm).
Response and Revision: Thank you very much for your meticulous comments and kind suggestions. Your advice has greatly benefited us. Your understanding is very rigorous. The SSR markers we detected were strongly linked to these genes, but can’t simply link a known gene with a marker. In subsequent studies, we will improve these genes by cryptographic linkage markers or sequence them in transgenic line.
Discussion
17、I haven’t read the entire discussion section but I don’t think the statement in lines 424-428 is correct for the reasons that I mentioned above. If quality traits were changed, how can you be sure that those changes are not negatively affecting health after consumption?
Response and Revision: Thank you very much for your meticulous comments and kind suggestions. Your advice has greatly benefited us. We added the transcriptomic and proteomic analyses in discuss. As yield traits are relatively easy to observe in backcross breeding, while quality traits cannot be observed, leading to the loss of some traits, according to your suggestion, we will not discuss the influence of genetic background on chalkiness for now. This section and its references removed from the discussion.
Methods
18、Combine sections 4.1, 4.2, 4.4, and 4.8. It should be combined using a heading like ‘Plante material and phenotyping’ and reorganized for better readability. Provide details on C.suppressalis evaluation with citation (if necessary)– the method for rearing the insect, infestation method, data recording (rating), etc. The same is true for agronomic and quality traits. One thing that bossers me is the number of environments (site x year) used for the field experiments. Most traits are quantitative that require at least 3 environments to get a better understanding of the effects of genotype by environment interaction. Such data are also required to compute broad-sens heritability that tells readers if the traits are repeatable within each trial and among trials, which is completely missing in the manuscript. This thing should be specified (at least it should be indicated as a weakness in the discussion section).
Response and Revision: Thank you very much for your meticulous comments and kind suggestions. Your advice has greatly benefited us. We reorganized these sections of the language and included them in sections 4.1 “Plante material and phenotyping”. We added to the details on C.suppressalis evaluation with citation – the method for rearing the insect, infestation method, data recording (rating), etc. Thank you very much for your advice, we pointed out the shortcomings of this study in the discussion, we lack the number of environments test, resulting in the inability to get a better understanding of the effects of genotype by environment interaction.
19、Sections 4.3, 4.5, 4.6, and 4.7 are for molecular techniques to understand variation at the DNA, RNA, and protein levels. The methodologies are well understood with numerous publications to cite. I suggest reducing this section by providing important information and citing appropriate references rather than writing everything as if this is the first study in the methodology.
Response and Revision: Thank you very much for your meticulous comments and kind suggestions. Your advice has greatly benefited us. We refined the content of these sections, retained important information, and added appropriate references.
20、Section 4.9 should have separate paragraphs to summarize how each of the phenotype data, genotype data, transcriptomics data, and protein data were analyzed with appropriate citations. The result sections have p values, error bars, etc. but the authors did not indicate how they did all those things.
Response and Revision: Thank you very much for your meticulous comments and kind suggestions. Your advice has greatly benefited us. We divided Section 4.9 into 3 separate paragraphs of the phenotype data, genotype data and protein data. According to your suggestion, the error bar and a test for significance are contradictory, so we only remian the error line,and we removed “Data were analyzed by one-way ANOVA, and treatment means were compared using the least significant difference test at P = 0.05. In addition, we didn't transcriptome analysis.

Reviewer 2 Report
I have only concern. Although authors state that “if the inserted genes in transgenic plants were involved in plant metabolic pathways, they may cause plant phenotypic changes” and in other words the unintended effects. And in line with this, they have found that the DEPs in two CRY gene transgenic lines are involved in various metabolic pathways, which means that these transgenes alter the plant metabolism and thus the uninteneded phenotypic changes in the transgenic plants are due to transgenes and NOT (or may be also) because of the genetic background. Could you pls make it clearer?
Author Response
Dear Editors:
No.: 2035990
“Unintended effects of transgenic rice on grain yield and quality traits determined by quantitative proteomics” Thank you very much for your effort in handling our manuscript. We greatly appreciate the thoughtful and constructive comments from the reviewers and have revised the paper accordingly. In addition, we also added the latest author ranking and correspondence unit information in the manuscript. We use blue text in the document to highlight changes to the manuscript to help you see them better. Thank you for offering us the opportunity of revising the manuscript. Enclosed please find the detailed, itemized list of our responses to the comments of the reviewers and the changes we have made to the paper. With the incorporation of suggestions from reviewer, the manuscript has been obviously improved. We hope that the revised manuscript will satisfy the reviewers and that you will find the paper is now suitable for publication in Plants.
Thanks again for your timely consideration of our manuscript.
On behalf of the authors,
Xiaosong Peng
Comments to author:
Question:I have only concern. Although authors state that “if the inserted genes in transgenic plants were involved in plant metabolic pathways, they may cause plant phenotypic changes” and in other words the unintended effects. And in line with this, they have found that the DEPs in two CRY gene transgenic lines are involved in various metabolic pathways, which means that these transgenes alter the plant metabolism and thus the uninteneded phenotypic changes in the transgenic plants are due to transgenes and NOT (or may be also) because of the genetic background. Could you pls make it clearer?
Response and Revision: Thank you very much for your meticulous comments and kind suggestions. Your advice has greatly benefited us. On one hand, As compared with the recurrent parent, the three transgenic lines had significantly lower stem borer infestation without showing significant differences in most of the agronomic traits, yield components, and end-use quality traits, However, the brown rice yield, head rice rate, and grain length/width ratio under insect pressure may be significantly higher than that of the parent due to the improvement of insect resistance. Therefore, We rearranged the logical order (see results 2.2 Intended effects analysis). The four lines for their reaction to the insect was significantly analyzed, and then qPCR and ELISA were used to explain the effect of CRY1C and CRY2A gene expression on insect resistance of rice.
On the other hand, we found that the DEPs in two CRY gene transgenic lines are involved in various metabolic pathways, but we found that the DEPs in two CRY gene transgenic lines are involved in various metabolic pathways, but the three transgenic lines without showing significant differences in most of the agronomic traits. Our study showed that the internal difference in genetic background is much larger than the variation of plant proteome caused by the introduction of foreign genes by transgenic technology or cross breeding. Due to lack the number of environments test, Resulting in the inability to get a better understanding of the effects of genotype by environment interaction. The genetic stability and environmental adaptability of these transgenic lines will be further investigated in this study. We added it to the discussion section.

Round 2
Reviewer 1 Report
The revised version is slightly improved, but it still looks to me like a preliminary working draft. The authors seem to be rushing to get a very poorly written paper without even checking their results presented in the text, tables, figures, and supplementary files. One of the important results the authors try to sell without facts, for example, is the unintended effects based on 12 phenotypic traits (agronomic and end-use quality traits). The text says something, Figure 12 tells a different story, and Table S9 tells another story. Similarly, the differentially expressed proteins section is full of contradictions: Tables 1-2, Figures 6-11, and Tables 7-8. Table 1 summarized 605 upregulated and 528 down-regulated proteins. In Table S7, the total number of differentially expressed proteins is 367. Figure 6 summarized about 283 DEPs involved in metabolism, poorly characterized, Cellular processes and signaling, and information storage and processing. Information storage and processing were not even provided in Table S7. Readers should be able to understand the results clearly, which is not the case. Too many redundant results that look to me worthless without clarity.
I have tried to partly edit it quickly to make it readable but don’t have time to finish all sections. The authors are free to consider anything they think improves their manuscript and complete the remaining section. Please spend time editing it properly rather than doing it in a hurry that will embarrass everyone involved in the writeup, editing, and reviewing. I have added the main points that I think are more relevant in the discussion section, which should be expanded more by citing appropriate references. Go through the track changes, accept/amend my changes, and finalize the remaining sections to make them clear, consistent, and unambiguous.

Author Response
Dear Editors:
No.: 2035990
“Comparison of the phenotypic performance, molecular diversity, and proteomics in transgenic rice”Thank you very much for your effort in handling our manuscript. We greatly appreciate the thoughtful and constructive comments from the reviewers and have revised the paper accordingly. Thank you for offering us the opportunity of revising the manuscript. Please see the attachment and the changes we have made to the paper. With the incorporation of suggestions from reviewer, the manuscript has been obviously improved. We hope that the revised manuscript will satisfy the reviewers.
Thanks again for your timely consideration of our manuscript.
On behalf of the authors, Xiaosong Peng
